# FreqBlender: Enhancing DeepFake Detection by Blending Frequency Knowledge

**Hanzhe Li**[1,*]   **Jiaran Zhou**[1,*]   **Yuezun Li**[1,†]   **Baoyuan Wu**[2]   **Bin Li**[3]   **Junyu Dong**[1]

[1] School of Computer Science and Technology, Ocean University of China
[2] School of Data Science, The Chinese University of Hong Kong, Shenzhen, China
[3] Guangdong Provincial Key Laboratory of Intelligent Information Processing,
Shenzhen University, Shenzhen, China

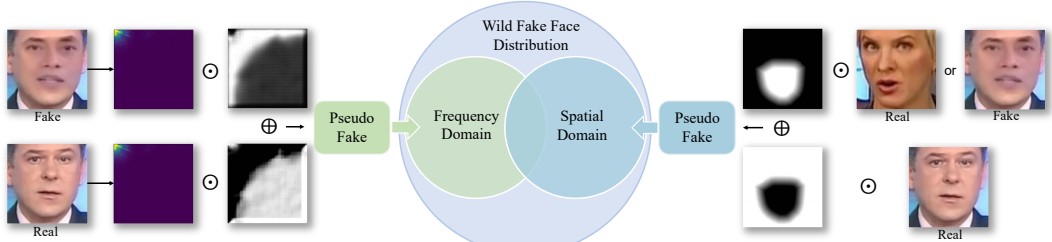

Figure 1: Overview of our method. In contrast to the existing spatial-blending methods (right part), our method explores face blending in frequency domain (left part). By leveraging the frequency knowledge, our method can generate pseudo-fake faces that closely resemble the distribution of wild fake faces. Our method can complement and work in conjunction with existing spatial-blending methods.

## Abstract

Generating synthetic fake faces, known as pseudo-fake faces, is an effective way to improve the generalization of DeepFake detection. Existing methods typically generate these faces by blending real or fake faces in spatial domain. While these methods have shown promise, they overlook the simulation of frequency distribution in pseudo-fake faces, limiting the learning of generic forgery traces in-depth. To address this, this paper introduces *FreqBlender*, a new method that can generate pseudo-fake faces by blending frequency knowledge. Concretely, we investigate the major frequency components and propose a Frequency Parsing Network to adaptively partition frequency components related to forgery traces. Then we blend this frequency knowledge from fake faces into real faces to generate pseudo-fake faces. Since there is no ground truth for frequency components, we describe a dedicated training strategy by leveraging the inner correlations among different frequency knowledge to instruct the learning process. Experimental results demonstrate the effectiveness of our method in enhancing DeepFake detection, making it a potential plug-and-play strategy for other methods.

## 1   Introduction

DeepFake refers to face forgery techniques that can manipulate facial attributes, such as identity, expression, and lip movement [1]. The recent advancement of deep generative models [2, 3] has

---

[*]Equal contribution
[†]Corresponds to Yuezun Li (liyuezun@ouc.edu.cn)

38th Conference on Neural Information Processing Systems (NeurIPS 2024).

greatly sped up the evolution of DeepFake techniques, enabling the creation of highly realistic and visually imperceptible manipulations. However, the misuse of these techniques can pose serious security concerns [4], making DeepFake detection more pressing than ever before.

There have been many methods proposed for detecting DeepFakes, showing their effectiveness on public datasets [5, 6, 7, 8, 9]. However, with the continuous growth of AI techniques, new types of forgeries constantly emerge, posing a challenge for current detectors to accurately expose unknown forgeries. To address this challenge, recent efforts [10, 11, 12, 13] have focused on improving the generalizability of detection, *i.e.*, the ability to detect unknown forgeries based on known examples. One effective approach to address this problem is to enhance the training data by generating synthetic fake faces, known as *pseudo-fakes* [14, 15, 16]. The intuition behind this approach is that the DeepFake generation process introduces artifacts in the step of blending faces, and these methods generate pseudo-fake faces by simulating various blending artifacts. By training on these pseudo-fake faces, the models can be driven to learn corresponding artifacts. However, existing methods concentrate on simulating the **spatial** aspects of face blending (see Fig. 1 (right)). While they can make the pseudo-fake faces resemble the distribution of wild fake faces in the spatial domain, they do not explore the distribution in the **frequency** domain. Thus, current pseudo-fake faces lack frequency-based forgery clues, limiting the models to learn generic forgery features.

In this paper, we shift our attention from the spatial domain to the frequency domain and propose a new method called *FreqBlender* to generate pseudo-fake faces by blending frequency knowledge (see Fig. 1 (left)). To achieve this, we analyze the composition of the frequency domain and accurately identify the range of forgery clues falling into. Then we replace this range of real faces with the corresponding range of fake faces to generate pseudo-fake faces. However, identifying the frequency range of forgery clues is challenging due to two main reasons: 1) this range varies across different fake faces due to its high dependence on face content, and 2) forgery clues may not be concentrated on a single frequency range but could be an aggregation of various portions across multiple ranges. Thus, general low-pass, high-pass, or band-pass filters are incapable of precisely pinpointing the distribution.

To address this challenge, we propose a Frequency Parsing Network (FPNet) that can adaptively partition the frequency domain based on the input faces. Specifically, we hypothesize that the faces are composed of three frequency knowledge, which represents *semantic information, structural information, and noise information*, respectively, and the forgery traces are likely hidden in structural information. This hypothesis is validated in our preliminary analysis (refer to Sec. 3 for details). Based on this footstone, we design the network consisting of a shared encoder and three decoders to extract corresponding frequency knowledge. The encoder transforms the input data into a latent frequency representation, while the decoders estimate the probability map of the corresponding frequency knowledge.

Training this network is non-trivial since no ground truth of frequency distribution is provided. Therefore, we propose a novel training strategy that leverages the inner correlations among different frequency knowledge. To be specific, we describe dedicated-crafted objectives that are performed on various blending combinations of the output from each decoder and emphasize the properties of each frequency knowledge. The experimental results demonstrate that the network successfully parses the desired frequency knowledge within the proposed training strategy.

Once the network is trained, we can parse the frequency component corresponding to the structural information of a fake face, and blend it with a real face to generate a pseudo-fake face. It is important to note that our method is not in conflict with existing spatial blending methods, but rather complements them by addressing the defect in the frequency domain. Our method is validated on multiple recent DeepFake datasets (*e.g.*, FF++ [5], CDF [6], DFDC [8], DFDCP [7], FFIW [9]) and compared with many state-of-the-art methods, demonstrating the efficacy of our method in improving detection performance.

The contributions of this paper are summarized in three-fold: **1)** To the best of our knowledge, we are the first to generate pseudo-fake faces by blending frequency knowledge. Our method pushes pseudo-fake faces closer to the distribution of wild fake faces, enhancing the learning of generic forgery features in DeepFake detection.**2)** We propose a Frequency Parsing Network that can adaptively partition the frequency components corresponding to semantic information, structural information, and noise information, respectively. Since no ground truth is provided, we design dedicated objectives

to train this network. **3)** Extensive experimental results on several DeepFake datasets demonstrate the efficacy of our method and its potential as a plug-and-play strategy for existing methods.

## 2 Related Works

The rapid progress of AI generative models has spawned the development of DeepFake detection methods. These methods mainly rely on deep neural networks to identify the inconsistency between real and fake faces using various features, including biological signals [17], spatial artifacts [18, 14, 15, 16], frequency abnormality [13, 19, 20], auto-learned clues from dedicated-designed models [21, 22]. These methods have shown promising results on public datasets. However, some of their performance significantly deteriorates when confronted with unknown DeepFake faces due to the large distribution discrepancy resulting from limited training datasets. To tackle this issue, many methods have been proposed to improve their generalizability by learning the generic DeepFake traces, *e.g.*, [10, 14, 15, 16, 12, 23, 24]. One effective approach is to create synthetic fake faces during training, known as pseudo-fake faces, *e.g.*, [10, 14, 15, 16]. FWA [10] is a pioneering method that conducts self-blending to simulate fake faces. Several extended variants (Face X-ray [14], PCL [15], SBI [16], BiG-Arts [25]) have been proposed to blend faces using curated strategies, further improving detection performance. By increasing the diversity of training faces, the gap in the distribution of wild fake faces can be reduced, allowing the models to learn the invariant DeepFake traces across different distributions. To generate the pseudo-fake faces, existing methods usually design spatial blending operations to combine different faces. This involves extracting the face region from a source image and blending it into a target image. However, these methods overlook the distribution of wild fake faces in the frequency domain. While the synthetic faces may resemble the spatial-based distribution, the lack of consideration for frequency perspective hinders the models from learning the fundamental generic DeepFake traces.

## 3 Preliminary Analysis

We perform a statistical analysis of the frequency distribution of real and fake faces and present preliminary results for the main frequency components corresponding to semantic information, structural formation, and noise information, respectively.

**Inspiration and Verification.** The investigation in previous works [26, 27] has indicated that the forgery traces mainly exist in high-frequency areas. However, the precise range of these areas has not been described, driving us to re-investigate the frequency distribution of forgery traces.

Specifically, we conduct verification experiments using FaceForensics++ (FF++) [5] datasets. We extract the frames from all videos and randomly select $3,000$ real images and $3,000$ fake images for each manipulation method (*e.g.*, DF, F2F, FS, and NT). Then we crop out the face region in these selected images using a face detector [28] and apply DCT [29] to generate frequency maps. For analysis, we sum up all frequency maps of real and fake images and adopt the visualization process of azimuthal average described in previous work [30, 27]. This process involves logarithmic transformation and the calculation of azimuthally-averaged flux in circular annuli apertures. By placing the center of the circular annuli aperture at the top-left corner of the frequency map, we can obtain a one-dimensional array representing the spectrum diagram. The visual results of their distribution are shown in Fig. 2 (top). It can be observed that this figure is consistent with the results in [26]. However, when we directly plot their distribution differences without logarithmic operation, the results do not match the previous figure. It can be seen that the disparities in high-frequency regions are not as substantial as expected, while the differences in the lower range become more noticeable (see Fig. 2 (bottom)). This is because the logarithmic operation mitigates the degree of differences in lower frequency ranges, causing the illusion that only the high-frequency range exhibits differences between real and fake faces. Therefore, we conjecture that the forgery traces may not only be concentrated in a very high-frequency range but could possibly spread to the low-frequency range.

**Hypothesis and Validation.** As shown in Fig. 2, the most significant difference can be observed in the range of very low frequency. Given the significant dissimilarity in appearance between real and fake faces, we hypothesize that the semantic information is mainly represented in this low-frequency band. Moreover, we hypothesize that the mid-to-high frequency components capture the structural information, making them more susceptible to containing forgery traces. Furthermore, we hypothesize

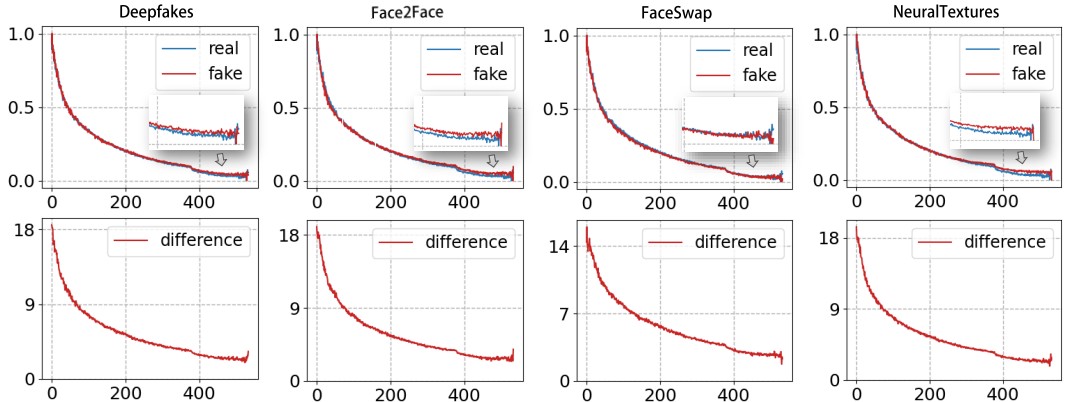

Figure 2: Statistics of frequency distribution. The top part shows the frequency distribution of real and fake faces using algorithms in [30, 27]. The bottom part shows the frequency difference between real and fake. The values on the vertical axis are logarithmic with 2.

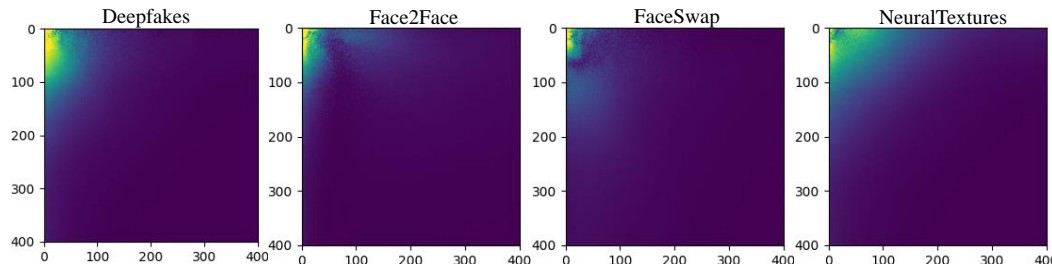

Figure 3: Visualization of the frequency difference between real and fake faces. The lighter color indicates the larger difference.

that the highest frequency components likely correspond to the noise introduced by various video preprocessing operations, such as compression, decompression, and encoding.

To validate our hypothesis, we directly visualize the difference between real and fake faces on their frequency maps in Fig. 3. By observing these results, we empirically split the frequency map into three non-overlap bands. The split operations follow the general band-pass filters. Denote the position in the frequency map as $(x, y)$, where $(0, 0), (1, 1)$ denotes the top-left corner and bottom-right corner. Specifically, we identify the region where $x + y \leq 1/16$ as containing semantic information, the region where $1/16 < x + y \leq 1/2$ as containing structural information, and the region where $x + y > 1/2$ as containing noise information. The corresponding results are visualized in Fig. 4, validating that these three ranges provide empirical evidence that aligns with our frequency distribution hypothesis.

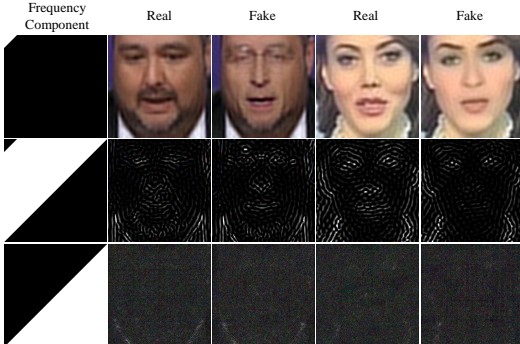

Figure 4: Image visualization corresponding to different frequency components.

## 4 FreqBlender

We describe a new method to create pseudo-fakes by blending specific frequency knowledge. The motivation is that existing methods only focus on spatial domain blending, which overlook the disparity between real and fake faces in the frequency domain. By considering the frequency distribution, the pseudo-fakes can closely resemble the fake faces. To achieve this, we propose

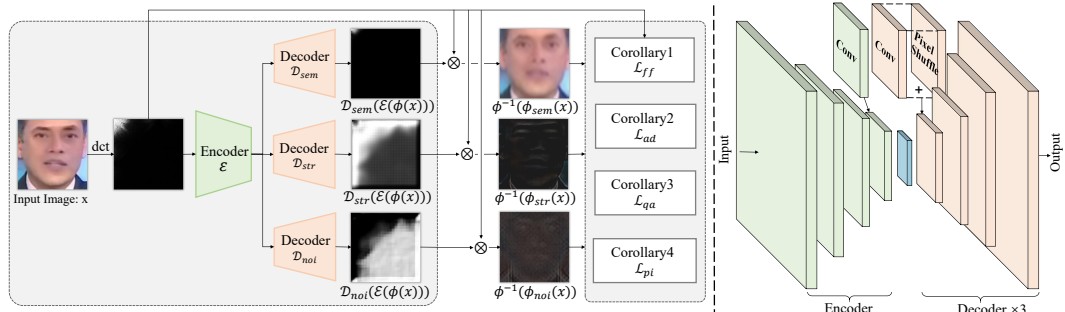

Figure 5: Overview of the proposed Frequency Parsing Network (FPNet). Given an input face image, our method can partition it into three frequency components, corresponding to the semantic information, structural information, and noise information respectively. Since there is no ground truth, we propose four corollaries to supervise the training. The architecture of the encoder and decoders is shown in the right part.

a Frequency Parsing Network (FPNet) to partition the frequency domain into three components, corresponding to semantic information, structural information, and noise information, respectively. We then blend the structural information of fake faces with the real faces to generate the pseudo-fakes. The details of the Frequency Parsing Network are elaborated in Sec. 4.1, and the objective and training process for this network is described in Sec. 4.2. Then we introduce the deployment of our method with existing methods in Sec. 4.3.

## 4.1 Frequency Parsing Network

**Overview.** The Frequency Parsing Network (FPNet) is composed of one shared encoder and three independent decoders. The encoder transforms the input faces into frequency-critical features and the decoders aim to decompose the feature from the encoder and extract the respective frequency components.

Denote the encoder as $\mathcal{E}$ and three decoders as $\mathcal{D}_{\mathrm{sem}}, \mathcal{D}_{\mathrm{str}}, \mathcal{D}_{\mathrm{noi}}$ respectively. Given an input face image $\mathbf{x} \in \mathcal{X} \in \{0, 255\}^{h \times w \times 3}$, we first convert this face to the frequency map as $\phi(\mathbf{x}) \in \mathbb{R}^{h \times w \times 3}$, where $\phi$ denotes the operations of Discrete Cosine Transform (DCT). Then we send this frequency map into model and generate three distribution maps as $\mathcal{D}_{\mathrm{sem}}(\mathcal{E}(\phi(\mathbf{x}))) \in [0, 1]^{h \times w}, \mathcal{D}_{\mathrm{str}}(\mathcal{E}(\phi(\mathbf{x}))) \in [0, 1]^{h \times w}$, and $\mathcal{D}_{\mathrm{noi}}(\mathcal{E}(\phi(\mathbf{x}))) \in [0, 1]^{h \times w}$ respectively. Each distribution map indicates the probability of the corresponding frequency component distributed in the frequency map. Given these distribution maps, we can select the corresponding frequency components conveniently. For example, the frequency component corresponding to semantic information can be selected by $\phi_{\mathrm{sem}}(\mathbf{x}) = \phi(\mathbf{x})\mathcal{D}_{\mathrm{sem}}(\mathcal{E}(\phi(\mathbf{x})))$ and the same is for other two frequency components, *i.e.*, $\phi_{\mathrm{str}}(\mathbf{x}) = \phi(\mathbf{x})\mathcal{D}_{\mathrm{str}}(\mathcal{E}(\phi(\mathbf{x})))$ and $\phi_{\mathrm{noi}}(\mathbf{x}) = \phi(\mathbf{x})\mathcal{D}_{\mathrm{noi}}(\mathcal{E}(\phi(\mathbf{x})))$. The overview of FPNet is shown in Fig. 5 (left).

**Network Architecture.** The encoder simply consists of four convolution layers with a kernel size of $3 \times 3$, a stride of 2, and a padding of 1. Each decoder also consists of four layers, and each layer is a combination of a convolutional layer and PixelShuffle operation [31] (see Fig. 5 (right)).

## 4.2 Objective Design for FPNet

The most challenging and crucial aspect of our method is to train the network for frequency parsing, as there is no ground truth available for the different frequency components. Note that the only available resources for supervising the training are the preliminary analysis results in Section 3. Nevertheless, these results are not precise and can not be adaptive to different inputs, which are insufficient for model training. Therefore, we meticulously craft a couple of auxiliary objectives to instruct the learning of networks, allowing for the self-refinement of the network.

These objectives are designed based on the following proposition.

**Proposition 1.** *Each frequency component exhibits the following properties:*

  *1. Semantic information can reflect the facial identity.*

2. *Structural information serves as the carrier of forgery traces.*

3. *Noise information has minimal impact on visual quality.*

4. *The preliminary analysis findings are generally applicable.*

**Corollary 1.** *For a given face $\mathbf{x}$, the transformed face based on its semantic information will retain the same facial identity as $\mathbf{x}$, i.e., $\forall \mathbf{x} \in \mathcal{X} \in \{0, 255\}^{h \times w \times 3}$, $\mathcal{F}(\phi^{-1}(\phi_{\mathrm{sem}}(\mathbf{x}))) = \mathcal{F}(\mathbf{x})$, where $\mathcal{F}$ denotes a face recognition model and $\phi^{-1}$ denotes the Inverse Discrete Cosine Transform (IDCT).*

**Facial Fidelity Loss.** We introduce a facial fidelity loss $\mathcal{L}_{\mathrm{ff}}$ to penalize the discrepancy in identity between the input face image and the spatial content represented by semantic information. To measure the identity discrepancy, we employ the MobileNet [32] as our face recognition model and train it using ArcFace [33, 34]. We select MobileNet for its balance between computational efficiency and recognition accuracy. Let $\mathcal{F}$ be the MobileNet and $\mathcal{F}_f(\mathbf{x})$ be the facial features extracted from the face image $\mathbf{x}$. The facial fidelity loss can be defined as

$$\mathcal{L}_{\mathrm{ff}}(\mathbf{x}) = \|\mathcal{F}_f(\phi^{-1}(\phi_{\mathrm{sem}}(\mathbf{x}))) - \mathcal{F}_f(\mathbf{x})\|_2^2. \tag{1}$$

Note that the input face $\mathbf{x}$ can be either real or fake, as the identity information is present in both cases.

**Corollary 2.** *For a given real face $\mathbf{x}_r$, it can be detected as fake if and only if it is inserted the structural information from a fake face $\mathbf{x}_f$, i.e., $\mathcal{D}(\mathbf{x}_r) = 0$ iff $\mathbf{x}_r \leftarrow \mathbf{x}_r \oplus \phi_{\mathrm{str}}(\mathbf{x}_f)$, where $\mathcal{D}$ denotes a Deepfake detector with labels of fake and real in $\{0, 1\}$, $\oplus$ indicates the inserting operation.*

**Authenticity-determinative Loss.** This loss is designed to emphasize the determinative role of structural information. To evaluate the authenticity of faces, we develop a DeepFake detector $\mathcal{D}$, which is implemented using a ResNet-34 [35] trained on real and fake faces. Then we construct two sets of faces by blending frequency components.

The first set contains three types of faces transformed from frequency components corresponding to 1) the semantic information of the real face, 2) the semantic information of the fake face, and 3) the semantic information of the real face blended with the structural information of the real face. We denote this set as $\mathcal{C}_r = \{\phi^{-1}(\phi_{\mathrm{sem}}(\mathbf{x}_r)), \phi^{-1}(\phi_{\mathrm{sem}}(\mathbf{x}_f)), \phi^{-1}(\phi_{\mathrm{sem}}(\mathbf{x}_r) + \phi_{\mathrm{str}}(\mathbf{x}_r))\}$. Since there is no structural information from fake faces in this set, all the faces should be detected as real.

Similarly, the second set contains two types of faces: 1) blending the semantic information of the fake face with the structural information of the fake face, and 2) blending the semantic information of the real face with the structural information of the fake face. We denote this set as $\mathcal{C}_f = \{\phi^{-1}(\phi_{\mathrm{sem}}(\mathbf{x}_f) + \phi_{\mathrm{str}}(\mathbf{x}_f)), \phi^{-1}(\phi_{\mathrm{sem}}(\mathbf{x}_r) + \phi_{\mathrm{str}}(\mathbf{x}_f))\}$. Since all blended faces in this set contain the structural information of fake faces, they should be detected as fake. Thus the authenticity-determinative loss $\mathcal{L}_{\mathrm{ad}}$ can be written as

$$\mathcal{L}_{\mathrm{ad}}(\mathbf{x}_r, \mathbf{x}_f) = \frac{1}{|\mathcal{C}_r|} \sum_{\mathbf{x} \in \mathcal{C}_r} \mathrm{CE}(\mathbf{x}, 1) + \frac{1}{|\mathcal{C}_f|} \sum_{\mathbf{x} \in \mathcal{C}_f} \mathrm{CE}(\mathbf{x}, 0) \tag{2}$$

where CE denotes the cross-entropy loss.

**Corollary 3.** *The face should exhibit no visible change if the frequency component of noise information is removed, i.e., $\forall \mathbf{x} \in \mathcal{X} \in \{0, 255\}^{h \times w \times 3}$, $\mathbf{x} \approx \mathbf{x} \ominus \phi_{\mathrm{noi}}(\mathbf{x}_f)$, where $\ominus$ indicates the removing operation.*

**Quality-agnostic Loss.** As noise information does not contain decisive details for the overall depiction of the image, the face image is expected to be similar to the face image transformed using the frequency components of semantic and structural information. This similarity can be quantified using the quality-agnostic Loss $\mathcal{L}_{\mathrm{qa}}$, defined as

$$\mathcal{L}_{\mathrm{qa}}(\mathbf{x}) = \|\mathbf{x} - \phi^{-1}(\phi_{\mathrm{sem}}(\mathbf{x}) + \phi_{\mathrm{str}}(\mathbf{x}))\|_2^2, \tag{3}$$

where the face $\mathbf{x}$ can be either real or fake.

**Corollary 4.** *Each frequency component is bound by the preliminary results, i.e., there should be no significant deviation between the predicted frequency component and the approximate frequency distribution in preliminary analysis.*

**Prior and Integrity Loss.** According to our analysis in the Preliminary Analysis section, we have an initial understanding of the approximate frequency distribution. Denote the initial frequency maps for semantic, structural, and noise information as $m_{\text{sem}}, m_{\text{str}}, m_{\text{noi}}$, respectively. These maps are utilized to accelerate the convergence of the model towards the desired direction. Moreover, we add a constraint on the integrity of their distributions, ensuring that their combination covers all elements of the frequency map. This loss $\mathcal{L}_{\text{pi}}$ can be expressed as

$$\mathcal{L}_{\text{pi}} = \|\mathcal{D}_{\text{sem}}(\mathcal{E}(\phi(\mathbf{x}))) - m_{\text{sem}}\|_2^2 + \|\mathcal{D}_{\text{str}}(\mathcal{E}(\phi(\mathbf{x}))) - m_{\text{str}}\|_2^2 + \|\mathcal{D}_{\text{noi}}(\mathcal{E}(\phi(\mathbf{x}))) - m_{\text{noi}}\|_2^2 +$$
$$\|(\mathcal{D}_{\text{sem}}(\mathcal{E}(\phi(\mathbf{x}))) + \mathcal{D}_{\text{str}}(\mathcal{E}(\phi(\mathbf{x}))) + \mathcal{D}_{\text{noi}}(\mathcal{E}(\phi(\mathbf{x})))) - \mathbf{1}\|_2^2,$$

(4)

where $\mathbf{1}$ denotes a mask where all the elements in it is $1$.

**Overall Objectives.** The overall objectives are the summation of all these loss terms, as

$$\mathcal{L} = \lambda_1 \mathcal{L}_{\text{ff}} + \lambda_2 \mathcal{L}_{\text{ad}} + \lambda_3 \mathcal{L}_{\text{qa}} + \lambda_4 \mathcal{L}_{\text{pi}},$$

(5)

where $\lambda_1, \lambda_2, \lambda_3, \lambda_4$ are the weights for different loss terms.

### 4.3 Deployment of FreqBlender

Given a fake face $\mathbf{x}_r$ and a real face $\mathbf{x}_f$, we can generate a pseudo-fake face by

$$\mathbf{x}_f' = \phi^{-1}\left(\phi(\mathbf{x}_r)\mathcal{D}_{\text{sem}}(\mathcal{E}(\phi(\mathbf{x}_f))) + \phi(\mathbf{x}_f)\mathcal{D}_{\text{str}}(\mathcal{E}(\phi(\mathbf{x}_f))) + \phi(\mathbf{x}_r)\mathcal{D}_{\text{noi}}(\mathcal{E}(\phi(\mathbf{x}_f)))\right).$$

(6)

◀) *Note that in our method, **it is not necessary to perform the blending using wild fake faces.** Instead, we can tactfully substitute wild fake faces with the pseudo-fake faces generated by existing spatial face blending methods. It allows us to overcome the limitations in the frequency distribution of existing pseudo-fake faces.*

## 5 Experiments

### 5.1 Experimental Setups

**DataSets.** Our method is evaluated using several standard datasets, including FaceForensics++ [5] (FF++), Celeb-DF (CDF) [6], DeepFake Detection Challenge (DFDC) [8], DeepFake Detection Challenge Preview (DFDCP) [7], and FFIW-10k (FFIW) [9] datasets. Specifically, the FF++ dataset consists of 1000 pristine videos and 4000 manipulated videos corresponding to four different manipulation methods, that are Deepfakes (DF), Face2Face (F2F), FaceSwap (FS), and NeuralTextures (NT). CDF dataset comprises 590 pristine videos and 5639 high-quality fake videos created from DeepFake alterations of celebrity videos available on YouTube. DFDC is a large-scale deepfake dataset, that consists of $100,000$ video clips, and DFDCP is a preview version of DFDC, which is also widely used in evaluation. The FFIW dataset contains 8250 pristine videos and 8250 DeepFake videos with multi-face scenarios. We follow the original training and testing split provided by the datasets for experiments.

**Implementation Details.** Our method is implemented using PyTorch 2.0.1 [36] with a Nvidia 3090ti. In the training stage of FPNet, the image size is set to $400 \times 400$. The batch size is set to 8 and the Adam optimizer is utilized with an initial learning rate of $1e^{-4}$. The training epoch is set to 200. The hyperparameters in the objective function in Eq. (5) are set as follows: $\lambda_1 = 1/12, \lambda_2 = 1, \lambda_3 = 1e^{-3}, \lambda_4 = 1/4$. For DeepFake detection, we employ the vanilla EfficientNet-b4 [37] as our model following [16]. In the training phase, we create pseudo-fake faces on-the-fly. We first generate synthetic faces using the spatial-blending method [16] and then blend them with real faces using our method with a probability of $\alpha = 0.2$. Other training and testing settings are the same as [16]. *More analysis of parameters are provided in Supplementary.*

### 5.2 Results

To showcase the effectiveness of our method, we train our method solely on the FF++ dataset and test it on the other different datasets. We employ the Area Under the Receiver Operating Characteristic Curve (AUC) as the evaluation metric following previous work [16]. Our method is compared with **five** video-based detection methods, including **Two-branch** [38], **DAM** [9], **LipForensics** [1],

Table 1: The cross-dataset evaluation of different methods. Blue indicates best and red indicates second best.

| Method | Input Type | Training Set | | Test Set AUC (%) | | | |
|---|---|---|---|---|---|---|---|
| | | Real | Fake | CDF | DFDC | DFDCP | FFIW |
| Two-branch (ECCV'20) [38] | Video | ✓ | ✓ | 76.65 | - | - | - |
| DAM (CVPR'21) [9] | Video | ✓ | ✓ | 75.3 | - | 72.8 | - |
| LipForensics (CVPR'21) [1] | Video | ✓ | ✓ | 82.4 | 73.50 | - | - |
| FTCN (ICCV'21) [39] | Video | ✓ | ✓ | 86.9 | 71.00 | 74.0 | 74.47 |
| SST (CVPR'24) [24] | Video | ✓ | ✓ | 89.0 | - | - | - |
| DSP-FWA (CVPRW'19 [10]) | Frame | ✓ | ✓ | 69.30 | - | - | - |
| Face X-ray (CVPR'20) [14] | Frame | ✓ | - | - | - | 71.15 | - |
| Face X-ray (CVPR'20) [14] | Frame | ✓ | ✓ | - | - | 80.92 | - |
| F3-Net (ECCV'20) [29] | Frame | ✓ | ✓ | 72.93 | 61.16 | 81.96 | 61.58 |
| LRL (AAAI'21) [40] | Frame | ✓ | ✓ | 78.26 | - | 76.53 | - |
| FRDM (CVPR'21) [41] | Frame | ✓ | ✓ | 79.4 | - | 79.7 | - |
| PCL+I2G (ICCV'21) [15] | Frame | ✓ | - | 90.03 | 67.52 | 74.37 | - |
| DCL (AAAI'22) [42] | Frame | ✓ | ✓ | 82.30 | - | 76.71 | 71.14 |
| SBI* (CVPR'22) [16] | Frame | ✓ | - | 92.94 | 72.08 | 85.51 | 85.99 |
| SBI (CVPR'22) [16] | Frame | ✓ | - | 93.18 | 72.42 | 86.15 | 84.83 |
| TALL-Swin (ICCV'23) [22] | Frame | ✓ | ✓ | 90.79 | 76.78 | - | - |
| UCF (ICCV'23) [12] | Frame | ✓ | ✓ | 82.4 | 80.5 | - | - |
| BiG-Arts (PR'23) [25] | Frame | ✓ | ✓ | 77.04 | - | 80.48 | - |
| F-G (CVPR'24) [43] | Frame | ✓ | ✓ | 74.42 | 61.47 | - | - |
| LSDA (CVPR'24) [23] | Frame | ✓ | ✓ | 83.0 | 73.6 | 81.5 | - |
| **FreqBlender (Ours)** | Frame | ✓ | - | **94.59** | 74.59 | **87.56** | **86.14** |

**FTCN** [39] and **SST** [24]. Moreover, we involve **thirteen** frame-level state-of-the-art methods for comparison, which are **DSP-FWA** [10], **Face X-ray** [14], F3-Net [29], **LRL** [40], **FRDM** [41], **PCL** [15], **DCL** [42], **SBI** [16], **TALL-Swin** [22], **UCF** [12], **SST** [24], **F-G** [43], **LSDA** [23],respectively.

**Cross-dataset Evaluation.** We evaluate the cross-dataset performance of our method compared to other counterparts in Table 1. The best performance is highlighted in blue and the second-best is marked by red. It should be noted that our method operates on pseudo-fake faces generated by SBI, thus we do not need fake faces. In comparison to video-level methods, our method achieves the best performance, which outperforms all the methods by a large margin.

When compared to frame-level methods, our method still outperforms the others. For example, our method improves upon the performance of the most relevant counterpart SBI by $1.41\%$, $2.17\%$, $1.41\%$, $1.31\%$ on CDF, DFDC, DFDCP, and FFIW respectively. This improvement can be attributed to the incorporation of frequency knowledge in pseudo-fake faces, enhancing the generalization of detection models. Note that the performance of the compared methods (except SBI* and F3-Net) is extracted from their original papers. SBI* denotes the performance obtained using the officially released codes, and F3-Net is reproduced using the codes implemented by others [3]. The results closely align with the reported scores, which verifies the correctness of our configuration of their codes. In subsequent experiments, we employ their release codes for comparison.

**Cross-manipulation Evaluation.** Since SBI is the most recent and effective method, we compare our method with it for demonstration. Specifically, we compare our method with two variants of the SBI method. The first is trained using the raw set of real videos in the FF++ dataset, while the second is trained using the c23 set. According to the standard protocols, all methods are tested on c23 videos. The results are shown in Table 2. It can be seen that our method outperforms SBI-raw by $3.62\%$ and SBI-c23 by $3.89\%$, demonstrating the efficacy of our method on cross-manipulation scenarios.

### 5.3 Analysis

**Effect of Each Objective Term.** This part studies the effect of each objective term on CDF, DFDC, and DFDCP datasets. The results are shown in Table 3. Note that Baseline denotes only using prior and integrity loss $\mathcal{L}_{\text{pi}}$ and "w/o" denotes without. It can be seen that without one certain objective

---
[3]F3-Net:https://github.com/Leminhbinh0209/F3Net

Table 2: The cross-manipulation evaluation of different methods.

| Method | FF++ | | | | Avg |
|---|---|---|---|---|---|
| | DF | F2F | FS | NT | |
| SBI-raw [16] | 98.35 | 91.07 | 96.92 | 83.69 | 92.51 |
| SBI-c23 [16] | 98.60 | 92.60 | 95.44 | 82.30 | 92.24 |
| **FreqBlender (Ours)** | 99.18 | 96.76 | 97.68 | 90.88 | **96.13** |

Table 3: Effect of each objective term.

| Setting | CDF | DFDC | DFDCP | Avg |
|---|---|---|---|---|
| Baseline | 91.69 | 72.69 | 86.67 | 83.68 |
| w/o $\mathcal{L}_{\mathrm{ff}}$ | 94.01 | 74.30 | 86.86 | 85.06 |
| w/o $\mathcal{L}_{\mathrm{ad}}$ | 93.31 | 72.97 | 86.32 | 84.20 |
| w/o $\mathcal{L}_{\mathrm{qa}}$ | 94.28 | 74.42 | 87.25 | 85.32 |
| w/o $\mathcal{L}_{\mathrm{pi}}$ | 93.78 | 74.03 | 85.99 | 84.60 |
| All | **94.59** | **74.59** | **87.56** | **85.58** |

Table 4: Effect of our method complementary to spatial-blending methods.

| Method | FF++ | CDF | DFDCP | FFIW | Avg |
|---|---|---|---|---|---|
| DSP-FWA [10] | 48.14 | 62.91 | 60.74 | 40.65 | 53.11 |
| DSP-FWA [10] + Ours | 49.46 | 65.47 | 56.18 | 41.81 | **53.23** |
| I2G [15] | 59.56 | 53.55 | 48.02 | 46.75 | 51.97 |
| I2G [15] + Ours | 63.84 | 48.89 | 49.53 | 48.96 | **52.81** |
| Face X-ray [14] | 82.26 | 67.99 | 65.00 | 63.65 | 69.73 |
| Face X-ray [14] + Ours | 84.03 | 76.05 | 63.90 | 67.24 | **72.81** |

Table 5: The effect of our method on different networks.

| Method | CDF | FF++ | DFDCP | FFIW | Avg |
|---|---|---|---|---|---|
| ResNet-50 [35] + SBI | 84.82 | 95.39 | 73.51 | 81.67 | 83.85 |
| ResNet-50 [35] + Ours | 85.44 | 94.61 | 76.16 | 86.32 | **85.63** |
| EfficientNet-b1 [37] + SBI | 90.25 | 94.66 | 87.54 | 82.55 | 88.75 |
| EfficientNet-b1 [37] + Ours | 90.53 | 94.65 | 87.70 | 83.76 | **89.16** |
| VGG16 [44] + SBI | 78.22 | 93.05 | 74.13 | 87.26 | **83.16** |
| VGG16 [44] + Ours | 78.38 | 93.10 | 73.47 | 87.63 | 83.15 |
| Xception [45] + SBI | 87.00 | 91.40 | 75.68 | 70.24 | 81.08 |
| Xception [45] + Ours | 90.52 | 93.32 | 76.07 | 70.43 | **82.59** |
| ViT [46] + SBI | 85.85 | 96.09 | 87.71 | 86.05 | 88.92 |
| ViT [46] + Ours | 86.34 | 96.10 | 87.17 | 86.88 | **89.12** |
| F3-Net [29] + SBI | 84.94 | 93.42 | 79.29 | 73.42 | 82.77 |
| F3-Net [29] + Ours | 88.10 | 95.16 | 84.32 | 74.49 | **85.52** |
| GFFD [41] + SBI | 81.34 | 91.81 | 77.19 | 65.53 | 78.97 |
| GFFD [41] + Ours | 86.71 | 92.18 | 78.25 | 77.45 | **83.65** |

term, the performance drops on all datasets, which demonstrates that different objective terms have distinct impacts, and their collective contributes most to our method.

**Complementary to Spatial-blending Methods.** To validate the complementary of our method, we replace the SBI method with other spatial-blending methods and study if the performance is improved. Specifically, we reproduce the pseudo-fake face generation operations in DSP-FWA, I2G, and Face X-ray, and combine them with our method. Note that I2G and Face X-ray have not released their codes, we re-implement them rigorously following their original settings. The results on FF++, CDF, DFDCP, FFIW datasets are presented in Table 4. It can be seen that by combining with DSP-FWA, our method improves $0.12\%$ averagely. A similar trend is also observed in the I2G, which improves $0.84\%$ on average. For Face X-ray, we improve the performance by $3.08\%$ on average. It is noteworthy that I2G and Face X-ray have not released their codes yet. We rigorously follow the instructions as in their papers and run the codes widely used by others[4][5].

**Different Network Architectures.** This part validates the effectiveness of our method on different networks, including ResNet-50 [35], EfficientNet-b1 [37], VGG16 [44], Xception [45], ViT [46], F3-Net [29], and GFFD[41]. We compare our method with SBI on these networks, which are tested on CDF, FF++, DFDCP, and FFIW datasets. The results are shown in Table 5. It can be observed that our method improves the performance by $1.78\%$, $0.41\%$, $1.51\%$, $0.2\%$, $2.75\%$, and $4.68\%$ averagely on ResNet-50, EfficientNet-b1, Xception, Vit networks, F3-Net, and GFFD networks respectively. It is noteworthy that our method slightly reduces the performance of VGG16 by $0.01\%$. It is possibly because the capacity of VGG16 is limited than other networks, and learning spatial pseudo-fake faces almost fills up this capacity, leaving no room for the learning of frequency knowledge.

---

[4]I2G: https://github.com/jtchen0528/PCL-I2G
[5]Face X-ray: https://github.com/AlgoHunt/Face-Xray

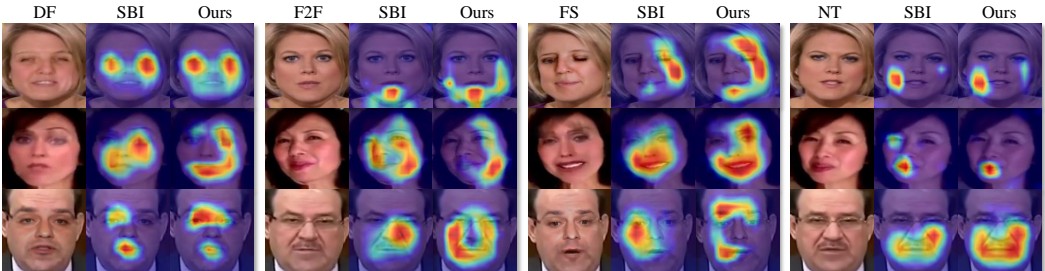

Figure 6: Grad-CAM visualization of SBI and our method on four manipulation types of FF++ dataset. Compared to SBI, our method focuses more on the manipulated structural boundaries.

Table 6: Effect of using wild fake or SP-fake faces.

| SP-fake | CDF | DFDC | DFDCP | FFIW | Avg |
|---------|-------|-------|-------|-------|-------|
| ✗ | 75.79 | 66.30 | 66.30 | 67.70 | 72.95 |
| ✓ | 94.59 | 74.59 | 74.59 | 86.14 | 85.72 |

**Saliency Visualization.** We employ Grad-CAM [47] to visualize the attention of our method compared to SBI on four manipulations in the FF++ dataset. Compared to SBI, our method concentrates more on the structural information, such as the manipulation boundaries. For example, our method highlights the face outline in DF, F2F, and FS, while focusing on the mouth contour in NT.

**Effect of Using Wild Fake or Spatial Pseudo-fake Faces.** As described in Sec. 4.3, our method is performed using real and spatial-blending pseudo-fake (SP-fake) faces. The rationale is that SP-fake faces are greatly diversified, containing more spatial forgery traces. Applying our method to these faces can consider both frequency and spatial traces effectively. To verify this, we directly perform our method on real and wild fake faces. The results in Table 6 indicate a notable performance drop when only wild fakes are used.

**Limitations.** Our method is designed to address the drawbacks of existing spatial-blending methods. Hence, it inherits the assumption that the faces are forged by face-swapping techniques. Further research is needed to validate our performance on other types of forgery operations, such as whole-face synthesis and attribute editing.

## 6 Conclusion

This paper describes a new method called *FreqBlender* that can generate pseudo-fake faces by blending frequency knowledge. To achieve this, we propose a Frequency Parsing Network that adaptively extracts the frequency component corresponding to structural information. Then we can blend this information from fake faces into real faces to create pseudo-fake faces. The extensive Experiments demonstrate the effectiveness of our method and can serve as a complementary module for existing spatial-blending methods.

**Acknowledgement.** This work is supported in part by the National Natural Science Foundation of China (No.62402464), Shandong Natural Science Foundation (No.ZR2024QF035), and China Postdoctoral Science Foundation (No.2021TQ0314; No.2021M703036). Jiaran Zhou is supported by the National Natural Science Foundation of China (No.62102380) and Shandong Natural Science Foundation (No.ZR2021QF095, No.ZR2024MF083). Baoyuan Wu is supported by Guangdong Basic and Applied Basic Research Foundation (No.2024B1515020095), National Natural Science Foundation of China (No. 62076213), and Shenzhen Science and Technology Program (No. RCYX20210609103057050). Bin Li is supported in part by NSFC (Grant U23B2022, U22B2047).

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

# A    Appendix / supplemental material

**Various Loss Weights** $\lambda_1, \lambda_2, \lambda_3, \lambda_4$. These weights are empirically selected based on experimental results. As shown in Table 7, we evaluate the performance across various set of $\lambda_1, \lambda_2, \lambda_3, \lambda_4$. The results exhibit that our method is not particularly sensitive to the settings of loss weights, with performance variations within approximately $\sim 0.5\%$. For our main experiments, we select the set of $\lambda_1 = 1/12, \lambda_2 = 1, \lambda_3 = 0.001, \lambda_4 = 1/4$ as it yields the best performance.

Table 7: Effect of 3ur method on different loss proportions.

| $\lambda_1$ | $\lambda_2$ | $\lambda_3$ | $\lambda_4$ | CDF | FF++ | FFIW | DFDCP | Avg |
|---|---|---|---|---|---|---|---|---|
| 1 | 1 | 1 | 1 | 93.16 | 96.30 | 87.82 | 85.32 | 90.65 |
| 0.1 | 1 | 0.1 | 0.5 | 93.59 | 95.60 | 84.78 | 86.60 | 90.14 |
| 0.1 | 1 | 0.01 | 0.5 | 94.27 | 96.11 | 85.54 | 87.81 | 90.93 |
| 0.01 | 1 | 0.0001 | 0.1 | 93.60 | 96.00 | 85.85 | 87.38 | 90.71 |
| 1/12 | 1 | 0.001 | 1/4 | 94.59 | 96.13 | 86.14 | 87.56 | 91.11 |

**More Details of Complementary to Spatial-blending Methods.** Table 8 shows the detailed results of every manipulation in the FF++ dataset. It can be seen that our method improves the performance of all manipulation methods, averaging $1.32\%$ for DSP-FWA, $4.28\%$ for I2G, and $1.77\%$ for Face X-ray. This improvement further demonstrates the effectiveness of our method.

Table 8: Effect of our method complementary to spatial-blending Methods.

| Method | FF++ | | | | |
|---|---|---|---|---|---|
| | DF | F2F | FS | NT | Avg |
| DSP-FWA [10] | 55.48 | 43.79 | 49.26 | 44.05 | 48.14 |
| DSP-FWA [10] + Ours | 56.20 | 45.93 | 53.96 | 41.74 | **49.46** |
| I2G [15] | 47.83 | 82.13 | 60.82 | 47.47 | 59.56 |
| I2G [15] + Ours | 56.90 | 81.26 | 61.66 | 55.52 | **63.84** |
| Face X-ray [14] | 89.38 | 85.02 | 83.45 | 71.17 | 82.26 |
| Face X-ray [14] + Ours | 93.21 | 85.41 | 82.70 | 74.79 | **84.03** |

**Probability $\alpha$ in FreqBlender.** As shown in Table 9, we evaluate the effect of using various probability $\alpha$. Note that $\alpha$ denotes the probability of generating a synthetic face whether using spatial-blending or using FreqBlender after spatial-blending. $\alpha = 1$ denotes only using FreqBlender, while $\alpha = 0$ means only using spatial-blending. It can be observed that solely using our method can not perform well, as no color space knowledge is involved, hindering the overall effectiveness of pseudo-fake faces. In contrast, solely using spatial-blending operations can reach a favorable performance. However, when inserting pseudo-fake faces generated by FreqBlender, the generalization performance is further enhanced, as our method can compensate for the loss of frequency knowledge. The optimal effect on CDF evaluation is achieved when $\alpha$ is set to 0.2. This experiment demonstrates the complementary effect of our method to existing spatial-blending methods. More generalization experiments on more datasets have been conducted in the main paper, demonstrating the effectiveness of our method.

Table 9: Effect of our method on different data proportions.

| $\alpha$ | FF++ | | | | | CDF |
|---|---|---|---|---|---|---|
| | DF | F2F | FS | NT | Avg | |
| 1 | 63.18 | 59.85 | 65.79 | 54.61 | 60.86 | 56.34 |
| 0.8 | 98.58 | 94.80 | 97.69 | 89.83 | 95.23 | 90.27 |
| 0.5 | 99.03 | 96.99 | 97.81 | 91.68 | 96.38 | 91.83 |
| 0.3 | 99.12 | 97.15 | 97.93 | 90.91 | 96.28 | 93.76 |
| 0.2 | 99.18 | 96.76 | 97.68 | 90.88 | 96.13 | **94.59** |
| 0.1 | 99.11 | 97.11 | 97.93 | 90.91 | 96.26 | 93.76 |
| 0 | 99.17 | 97.63 | 97.77 | 91.35 | 96.50 | 93.58 |

**Visual Demonstration of FreqBlender.** The goal of our method is to create pseudo-fake faces that resemble the frequency distribution of wild fake faces. To verify this, we conduct a visual experiment on the FF++ dataset. Specifically, we randomly select $3,000$ images from each manipulation method (Deepfakes, FaceSwap, Face2Face, NeuralTextures) and calculate the average frequency map for each manipulation method respectively. Then we create the same number of pseudo-fake faces using

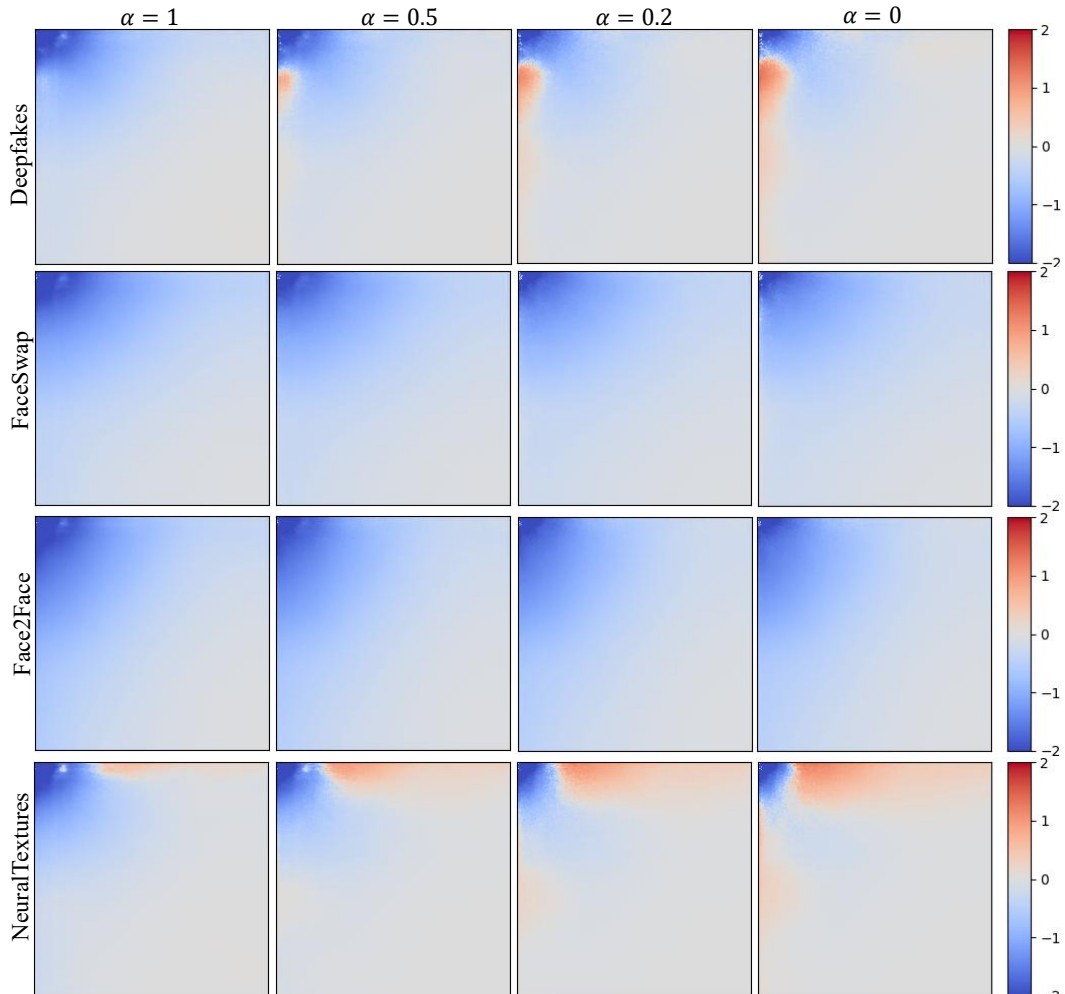

Figure 7: Heatmap visualization of the frequency difference between our method and the wild fake faces (Deepfakes, FaceSwap, Face2Face, NeuralTextures). Note that $\alpha = 0$ denotes that our method is degraded to SBI [16].

our method and calculate their average frequency map. Finally, we visualize the frequency difference between our method and wild fake faces. The results are illustrated in Fig. 7. It is important to note that $\alpha$ represents the probability of applying FreqBlender. Hence, $\alpha = 1$ means all pseudo-fake faces are created using our method, while $\alpha = 0$ denotes our method is degraded to SBI. From this figure, we can see that the difference is minimal when $\alpha = 1$. As $\alpha$ decreases, the difference becomes larger as the pseudo-fake faces are more likely created by SBI. This demonstrates that our method effectively simulates the frequency distribution of wild fake faces.

**Performance on FF++ Low-quality (LQ).** To investigate the performance of our method on low-quality videos, we conduct experiments on the FF++ Low-quality (LQ) set. The results in Table 10 show that while all methods experience substantial performance drops on the LQ set, our method consistently achieves the highest performance, demonstrating better generalization capability on low-quality videos compared to other methods.

**Tentative Validation on Various Synthesized Faces.** In addition to validation on the standard datasets, we investigate a new scenario: Diffusion-based face-swap deepfake detection. In this scenario, we employ a recent diffusion model (Collaborative Diffusion [48]) to synthesize faces, which are then blended into original videos. We create 200 fake faces and evaluate our method in Table 11. It can be seen that our method is effective in detecting such forged faces.

Table 10: Performance of our method on FF++ (LQ).

| Method | FF++(LQ) |
|---|---|
| I2G [15] | 52.20 |
| Face X-ray [14] | 65.41 |
| SBI [16] | 76.11 |
| FreqBlender | 77.56 |

Table 11: Performance of Diffusion-based face-swap and Gan-Generated images.

| Method | Diffusion-based [48] | StyleGAN [49] | StyleGAN2 [50] |
|---|---|---|---|
| I2G [15] | 63.51 | 47.89 | 43.86 |
| Face X-ray [14] | 89.81 | 59.11 | 66.54 |
| SBI [16] | 91.70 | 63.99 | 72.88 |
| FreqBlender | 94.74 | 64.39 | 76.70 |

Moreover, although our method is designed for face-swapping techniques, we also test it on face images generated by StyleGAN [49] and StyleGAN2 [50]. The results, also shown in Table 11, reveal a notable performance drop across all methods. However, our method still outperforms the others.

**Performance against Evasion Attacks.** We employ a widely recognized library `TorchAttacks` with four well-known attack methods: FGSM [51], BIM [52], PGD [53] and CW [54]. The attack experiment is conducted on the CDF dataset with six models and the attack configuration is set by default. The results are shown in Table 12. It can be seen without any defense strategies, all models can be easily attacked in the white-box attacking mode, which exactly aligns with our expectations and the discoveries in the papers of FGSM, BIM, etc.

Table 12: AUC (%) performance against evasion attacks.

| Method | Original | Attacked | | | |
|---|---|---|---|---|---|
| | | FGSM | BIM | PGD | CW |
| FWA [10] | 62.91 | 27.43 | 0 | 0 | 0 |
| I2G [15] | 53.35 | 3.26 | 0 | 0 | 0 |
| Face-Xray [14] | 67.99 | 54.66 | 0 | 0 | 0 |
| SBI-c23 [16] | 92.94 | 27.19 | 0 | 0 | 0 |
| SBI-raw [16] | 93.18 | 11.48 | 0 | 0 | 0 |
| Ours | 94.59 | 22.31 | 0 | 0 | 0 |

