# OpenReview forum: "FreqBlender: Enhancing DeepFake Detection by Blending Frequency Knowledge"
_NeurIPS.cc/2024/Conference — NeurIPS 2024 poster_

### Official Review · Reviewer_Z1eA · 2024-07-04

**Soundness:** 3
**Presentation:** 3
**Contribution:** 3
**Rating:** 6
**Confidence:** 5

**Summary:**

The generalization of DeepFake detection can be addressed by enhancing training data with synthetic fake faces, known as pseudo-fakes. Traditional methods generate these faces by spatial blending operations. However, the limitations of these methods are that they ignore to simulate the frequency distribution, where additional important forgery clues might be found.

This paper introduces an interesting method called FreqBlender, which attempts to blend proper frequency knowledge to enhance the effectiveness of pseudo-fake faces. This method involves identifying forgery clues in the frequency domain and blending the corresponding frequency components from fake faces into real faces. This process is challenging due to the variability and spread of forgery clues across different frequency ranges.

To achieve frequency decomposition, the authors propose a Frequency Parsing Network (FPNet) that can adaptively partition the frequency domain. The network, consisting of an encoder and three decoders, extracts semantic, structural, and noise information from the faces. Training FPNet is difficult due to the lack of ground truth for frequency distribution, so the authors devise a novel training strategy leveraging inner correlations among different frequency components.

Once trained, FPNet can extract the structural information of a fake face's frequency component and blend it with a real face to generate a pseudo-fake face. This method complements existing spatial blending methods and improves detection performance on multiple DeepFake datasets, demonstrating its effectiveness.

**Strengths:**

1. This papeer enhancing the generalization of DeepFake detection is a critical and current issue in AI safety, as real-world DeepFakes are likely generated by unknown models. This method addresses this challenge by taking an innovative approach, creating pseudo-fake faces through blending frequency knowledge rather than the conventional spatial knowledge used in existing methods.

2. This work identifies the limitations of existing methods, noting that spatial blending only mimics the visual similarity of wild fake faces but overlooks their frequency characteristics. By analyzing the frequency distribution, the authors describe a Frequency Parsing Network (FPNet) to parse the frequency range carrying forgery traces. Interestingly, this method doesn't require wild fake faces for blending; instead, it uses existing spatial-based pseudo-fake faces as surrogates.

6. The method's efficacy is validated across multiple recent DeepFake datasets, demonstrating its effectiveness with various backbones and complementing existing spatial-blending methods. This showcases its robustness and practical applicability in enhancing detection performance.

**Weaknesses:**

This paper proposes an interesting and intuitive method that seems reasonable to me. Here are some recommendations for future improvements:

1. Investigate more fine-grained frequency ranges to uncover the detailed composition of artifacts. This could potentially enhance the model's performance by providing a more nuanced understanding of forgery clues.

2. Develop algorithms to enhance the explainability of the model’s decisions. This will help users understand how and why a particular face is identified as a DeepFake, increasing trust and usability in practical applications.

3. Extend the method to handle other types of forgery operations beyond face-swapping, such as whole-face synthesis and attribute editing. This will broaden the method's applicability and ensure it remains effective across a wider range of DeepFake techniques.

4. Utilizing LLMs may help you parse the frequency more precisely, as LLMs can provide the prior knowledge which may compensate for the lack of ground truth when training FPNet.

**Questions:**

1.	As described in Section 3, the statistics of the frequency distribution are calculated using an azimuthal average, which includes a logarithmic transformation and the calculation of azimuthally-averaged flux in circular annular apertures. Could you provide more details on why placing the center of the circular annular aperture at the top-left corner of the frequency map results in a one-dimensional spectrum diagram?
2.	The FPNet uses three independent decoders to analyze the frequency domain. The output of the decoders is not clearly explained, as it is visualized in grayscale as the frequency map in Fig. 5. I understand that the output is a soft mask ranging from [0,1] for the frequency map. Therefore, I suggest using a color version to replace the DCT map in Fig. 5.
3.	In the discussion on the loss of authenticity determination, the authors aim to create two sets of faces, with and without forgery traces. Does the set C_r represent the face sets that do not contain structural information of fake faces, while C_f corresponds to the face sets that have structural information of fake faces?
4.	In Prior and Integrity Loss, the final term is designed to ensure that the combination of three frequency components covers the entire frequency domain. This term sums these frequency components and calculates the distance from a mask of 1. However, this term might not guarantee that the sum of these frequency components equals 1, but it can reduce deviations. What is the rationale behind this design??

**Limitations:**

This paper lists its limitations in main body. Since the proposed method is designed to address the limitations of existing spatial-blending techniques. it operates under the assumption that faces are forged using face-swapping methods. It unlikely tackle other types of forgeries, such as whole-face synthesis.

---

> ### Author Rebuttal · Authors · 2024-08-07
>
> Thank you for the positive review on the significance of our topic, novelty and experimental results.
>
>
> **Q1: Could you provide more details on why placing the center of the circular annular aperture at the top-left corner of the frequency map results in a one-dimensional spectrum diagram?**
>
> **R1:** Thanks for the question. We follow the process in [26,27], where we use the top-left corner of frequency map as the center for several circular annuli apertures. The radius of these apertures is varied from 0 up to the side length of the frequency map, with a fixed interval. Then we average the signals inside the interval area between adjacent circular annuli apertures. In this way, we convert a two-dimensional frequency map into a one-dimensional frequency spectrum. We will revise correpsonding descriptions to improve clarity.
>
> **Q2: I understand that the output is a soft mask ranging from [0,1] for the frequency map. Therefore, I suggest using a color version to replace the DCT map in Fig. 5.**
>
> **R2:** Thanks for your suggestion. We will replace it with a color version.
>
> **Q3: Does the set C_r represent the face sets that do not contain structural information of fake faces, while C_f corresponds to the face sets that have structural information of fake faces?**
>
> **R3:** Yes, your understanding is correct.
>
> **Q4: The Prior and Integrity Loss sums these frequency components and calculates the distance from a mask of 1. What is the rationale behind this design?**
>
> **R4:** Thanks for the insightful question. We expect these frequency components will span the entire frequency domain (all elements in frequency map) while minimizing overlap. Therefore, we calculate the distance of the sum of these frequency components using mask 1. We will enhance the clarity of the related descriptions in the revision.

---

> > ### Comment · Reviewer_Z1eA · 2024-08-12
> >
> > Thank you for the reply. The author responded to my concern directly, resolving my doubts. I believe this article has a positive significance for the community and the idea is quite novel, so I keep my score.

---

> > > ### Author Response · Authors · 2024-08-12
> > >
> > > Thanks for your comments! I am very glad to hear that our response addresses your concerns and resolves your doubts. We appreciate that the novelty of our idea is highly recognized and its positive significance to the community is well acknowledged.

---

### Official Review · Reviewer_GPf8 · 2024-07-12

**Soundness:** 3
**Presentation:** 3
**Contribution:** 3
**Rating:** 4
**Confidence:** 5

**Summary:**

This work studies generalizable deepfake detection. The proposed method is motivated by a new data augmentation method that blends real and fake faces in the frequency domain. The paper claims the forgery can be found in three different frequency bands and proposes an unsupervised learning method, Frequency Parsing Network. Empirical results indicate that the proposed method's performance achieves results comparable to state-of-the-art.

**Strengths:**

1. unsupervised way of learning the frequency component, which can favor its usage.
2. I like the analysis from Fig 2 and 3, showing the difference between real and fake residing in the high-frequency band.
3. I enjoy the problem formulation as well as the learning object in section 4.2.

**Weaknesses:**

1. the proposed method does not achieve state-of-the-art detection performance in table 1.
2. table 2 only has a few methods on the FF++ dataset. It should not be the case that SBI is only a baseline to compare.
3. the proposed approach might be limited in standard face-swap fake face, or gan-generated face. How about diffusion model generated face?
4. in terms of approach. the proposed method works when capturing semantic, structural and noise information. What if it fails to capture any one of those?

**Questions:**

1. line 31 says the motivation of simulating various blending artifacts. However, these artifacts are quite "old" compared to that from current diffusion models. That says, how would your method perform on the face generated by stable diffusion? such as instantid, photomaker?
2. did you compare with DIRE and HiFi-Net?
R1: DIRE for Diffusion-Generated Image Detection, ICCV2023
R2: Hierarchical fine-grained image forgery detection and localization, CVPR2023

**Limitations:**

I am impressed by the learning objective formulation, but I am inclined to reject it because it does not achieve reasonable sota performance and does not show generalization to diffusion-generated images.

---

> ### Author Rebuttal · Authors · 2024-08-07
>
> We sincerely thank the reviewer for the valuable time and comments.
>
> **Q1: The proposed method does not achieve state-of-the-art detection performance in table 1.**
>
> **R1:** We would like to highlight that **our method achieves the highest number of top-1 rankings compared to all others (best performance on 3 out of 4 datasets)**, and ranks 3rd on the DFDC dataset among the 20 methods compared. Given the wide variety of detection strategies and deepfake datasets, we believe this achievement can demonstrate the comprehensive superiority of our method.
>
> Moreover, we respectfully believe that both the AC and all reviewers could agree with that, the value and contribution of a work should not solely be judged by empirical results, but also by the innovative insights it offers for future research.
>
> **Q2: Table 2 only has a few methods on the FF++ dataset. It should not be the case that SBI is only a baseline to compare.**
>
> **R2:** The rationale for using SBI lies in two aspects:
> * **Second-best performance**: SBI consistently performs better than other methods (**coming in second only to our method, achieving the second-best performance on 2 out of 4 datasets**). Therefore, we believe that comparing our method with SBI is representative and can reflect the efficacy of our method in cross-manipulation evaluation.
> * **High relevance**: Our method adopts SBI as the substitute for fake faces in creating frequency-blending pseudo-fake faces. Thus, comparing our results with those of SBI further highlights the effectiveness of our method.
>
> We will include these explanations in the revision for better clarity.
>
>
> **Q3: The proposed approach might be limited in standard face-swap fake face, or gan-generated face. How about diffusion model generated face? This method does not show generalization to diffusion-generated images.**
>
> **R3:** Thanks for the thoughful question. We provide our responses in three aspects:
> * **Our method is in scope of face-swap deepfake detection**: As described in Introduction (L29-L31) and stated in the limitation section (L322-L325), our method targets for the face-swap deepfake detection, **a significant topic in recent years, as evidenced by works such as Face x-ray [14] (CVPR2020), PCL [15] (ICCV2021), SBI [16] (CVPR2022), UCF [12] (ICCV2023), BiG-Arts [25] (PR2023), F-G [43] (CVPR2024), and LSDA [23] (CVPR2024)**. This scope limitation has been noted by other reviewers as well.
> * **It is important to note that detecting diffusion-generated faces and face-swap deepfake are typically two different tasks**: Detecting Diffusion-generated images typically falls under the category of whole image synthesis, as seen in works such as DIRE (ICCV2023, the suggested work by reviewer), AVG (ICASSP2023). In these works, they do not validate themselves on face-swap deepfake datasets studied in our paper. Therefore, **while detecting diffusion-generated images is also an important task, it falls outside the scope of our method.**
> * **In response to the suggestion, we investigate whether our method can facilitate the detection of Diffusion-based generated images.** We conduct an additional scenario: **Diffusion-based face-swap deepfake detection**. In this scenario, a recent diffusion model (Collaborative Diffusion (CVPR2023)) is used to synthesize a face, which is then blended into original videos. We use the dataset provided by this work to create 200 fake faces, duo to the time contraints. **Table A** presents the performance of our method in this context, demonstrating its effectiveness in detecting such forged faces. We will include this experiment into the revision.
>
> **Table A: Performance of Diffusion-based face-swap deepfake detection.**
> |             | Diffusion-based  |
> |:-----------:|:----:      |
> | FreqBlender |     94.74     |
>
>
> **Q4: The proposed method works when capturing semantic, structural and noise information. What if it fails to capture any one of those?**
>
> **R4:** Thanks for the thoughtful question. The proposed method is capable of decomposing any input face into three frequency components. However, in certain exceptional cases (e.g., whole face synthsis or attribute editing), the captured structural information may not fully cover the forgery traces, affecting the effectiveness of pseudo-fake faces and subsequently limiting the efficacy of deepfake detectors.
>
> **Q5: "Line 31 says the motivation of simulating various blending artifacts. However, these artifacts are quite "old" compared to that from current diffusion models. How would your method perform on the face generated by stable diffusion?**
>
> **R5:** See response in R3.
>
> **Q6: Did you compare with DIRE and HiFi-Net?**
>
> **R6:** Thanks for the value question.
> * **DIRE is intended for detecting fully synthesized faces, rather than specifically identifying face-swap deepfakes.** To compare with ours, we adapt this method under our scenario. **Table B** illustrates the performance of both DIRE and our method, demonstrating that our method significantly outperforms DIRE.
> * **HiFi-Net, on the other hand, is built for general image manipulation localization (i.g., segmentation) and could not be used for face forgery detection.** Thus, HiFi-Net is not included in our comparisons.
>
> **Table B: Comparison with DIRE and HiFi-Net.**
> |             | CDF   | DFDC | DFDCP | FFIW  |
> |:-----------:|:-----:|:----:|:-----:|:-----:|
> | DIRE        | 46.71 | 51.97| 45.24 | 50.82 |
> | FreqBlender | 94.59 | 74.59| 87.56 | 86.14 |

---

> ### Author Response · Authors · 2024-08-12
>
> Dear Reviewer GPf8,
>
> We sincerely appreciate the time and thoughtful comments you’ve provided. With the remaining time being limited, we are eager to receive your feedback, especially on the issues we've addressed in our rebuttal.
>
> In our response, we have carefully reviewed your comments and provided the following summaries:
> 1. Clarified the concern regarding SOTA and its application to diffusion-based models.
> 2. Analyzed the suggested DIRE and HIFI-Net.
> 3. Addressed the questions about using SBI in Table 2 and the shortcomings of FPNet.
>
> We hope the new experiments and analysis have demonstrated the merits of our work. We deeply appreciate your time and effort!
>
> Best regards,
> Authors

---

> > ### Comment · Reviewer_GPf8 · 2024-08-14
> > **Review Comments**
> >
> > Thanks for the clarification from the authors. I am sorry that I mistakenly thought the method did not achieve the sota detection performance.
> >
> > However, I still keep my scores for the proposed method's limitation on FF++ and lack of convincing experiments to show the generalization ability to the diffusion face-swap.
> >
> > 1. FF++ is the core of deep fake detection, and cross-manipulation evaluation is common in the community [R1,R2,R3,R4,R5]. However, table 2 from the main table merely reports 2 methods, and NONE of these references are discussed in the submission. This is insufficient to conclude that the proposed method is effective enough.
> >
> > [R1] End-to-End Reconstruction-Classification Learning for Face Forgery Detection, CVPR2022
> >
> > [R2] Thinking in Frequency: Face Forgery Detection by Mining Frequency-aware Clues, ECCV2020
> >
> > [R3] UCF: Uncovering Common Features for Generalizable Deepfake Detection, ICCV2023
> >
> > [R4] Spatial-Phase Shallow Learning: Rethinking Face Forgery Detection in Frequency Domain, CVPR 2021
> >
> > [R5] Exploring Disentangled Content Information for Face Forgery Detection, ECCV 2022
> >
> > 2. Table 1 in the rebuttal is invalid, and I am not convinced by this experiment in **two** aspects:
> >
> > - no method was used for the comparison, only showing your performance is not enough.
> >
> > - why can't authors evaluate more commonly used diffusion-based methods when easily accessible tools are available,  such as stable 1.5, stable 2.1, instantiated, and Dalle2? Being selective on the face generation method is not fair. For example, [R6] reports the generalization performance on StarGAN, DDPM, DDIM, and SD.
> >
> > [R6] Transcending Forgery Specificity with Latent Space Augmentation for Generalizable Deepfake Detection, CVPR 2024
> >
> > 3. The proposed method is frequency-based, then what is the performance when the forgery trace largely occurs in the RGB domain whereas less on the frequency domain. For example, these cartoon faces with large eyes generated from SD-based method? will the performance decline?

---

> ### Author Response · Authors · 2024-08-14
>
> We sincerely appreciate your valuable time and additional comments you’ve provided.
>
> **Q1. FF++ is the core of deep fake detection, and cross-manipulation evaluation is common in the community [R1,R2,R3,R4,R5]. However, table 2 from the main table merely reports 2 methods, and NONE of these references are discussed in the submission. This is insufficient to conclude that the proposed method is effective enough.**
>
> **R1**. We would like to clarify that **the goal of our method is to create effective pseudo-fake faces solely using real faces, as in (Face X-ray [14], PCL [15], SBI [16])**. But the difference (novelty) is that we introduce FreqBlender to incorporates frequency information into these pseudo-fake faces.
>
>
> Typically, the efforts of this direction are trained on **real faces in FF++ without using fake faces.** This allows them to be fairly validated across all four tracks in FF++, demonstrating effectiveness in cross-manipulation scenarios. Therefore, we follow the protocol of these methods and compare our approach with theirs. **Since SBI shows the second-best performance, we limit our comparison to it in Table 2. However, additional studies involving more methods, including  [14,15,16]), are presented in Table 7 (Supplementary)**
>
> **Following your suggestion, we thoroughly review these papers and found that R2 and R4 do not conduct cross-manipulation evaluations, while R3 employs a less challenging scenario (training on three tracks and testing on one). Thus, R2, R3, and R4 are not suitable for direct comparison.**
>
> R1 and R5 are trained on the DF track of FF++. Comparing with these two methods are relative fair. The results, shown in **Table A**, highlight the notable superiority of our approach. We will include this comparison in the revision.
>
> **Table A: Cross-manipulation comparison.**
> |             | DF    | F2F  | FS    | NT    | Avg   |
> |:-----------:|:-----:|:----:|:-----:|:-----:|:-----:|
> | R1 (trained on DF)       | 99.65 |70.66 |74.29  |67.34  |77.99 |
> | R5 (trained on DF)         | 99.22 |60.18 |68.19  |61.17  |72.19 |
> |FreqBlender       | 99.18 |96.76 |97.68  |90.88  |96.13 |
>
>
> **Q2: Table 1 in the rebuttal is invalid, and I am not convinced by this experiment in two aspects: 1) no method was used for the comparison, only showing your performance is not enough. 2) why can't authors evaluate more commonly used diffusion-based methods when easily accessible tools are available, such as stable 1.5, stable 2.1, instantiated, and Dalle2? Being selective on the face generation method is not fair. For example, [R6] reports the generalization performance on StarGAN, DDPM, DDIM, and SD**
>
> **R2**: In the first round of rebuttal, we follow the suggestion to show the generalization of our method to diffusion models. The performance is 94.74, which we believe can demonstrate the efficacy. **As suggested, we evaluate more methods (I2G, Face X-ray, SBI) as in Table B**, which also demonstrate the efficacy of our method.
>
> To create diffusion-based face-swap deepfakes, **Collaborative Diffusion (CVPR 2023) is more user-friendly and efficient than Stable 1.5, Stable 2.1, Instantiated, and DALL-E 2**, as it allows for more effective editing of facial attributes compared to the suggested models.
>
>
> **Please note that evaluating on StarGAN, DDPM, DDIM, and SD is not the primary focus or contribution of R6, which is why its performance on these models is not satisfactory (around 73% on average)**. Following the suggestion, we have made an effort to validate our method on StyleGAN and StyleGAN2 (which provide ready-made face sets) and present the results in **Table C**. **Our method outperforms others but achieves performance comparable to R6**.
>
>
> **Table B: Performance of Diffusion-based face-swap deepfake detection.**
> |             | Diffusion-based  |
> |:-----------:|:----:      |
> | I2G |     63.51     |
> | Face X-ray |     89.81     |
> | SBI |    91.70     |
> | FreqBlender |     94.74     |
>
>
> **Table C: Results in Gan-generated images.**
> |             | StyleGan  |   StyleGan2   |
> |:-----------:|:---------:|:-------------:|
> | I2G         |  47.89    |    43.86      |
> | Face X-ray  |  59.11    |    66.54      |
> | SBI         |  63.99    |    72.88      |
> | FreqBlender |  64.39    |    76.70      |
>
> **Q3: The proposed method is frequency-based, then what is the performance when the forgery trace largely occurs in the RGB domain whereas less on the frequency domain. For example, these cartoon faces with large eyes generated from SD-based method? will the performance decline?**
>
> **R3**: We would like to emphasize that our method does not rely solely on frequency information. **As highlighted in L241, our approach integrates frequency knowledge into the existing spatial-blending pseudo-fake faces, allowing it to address both spatial and frequency aspects effectively.**
>
> We hope this explanation clarifies your concerns and encourages a re-evaluation of our work’s contribution.

---

### Official Review · Reviewer_Xqko · 2024-07-13

**Soundness:** 3
**Presentation:** 3
**Contribution:** 3
**Rating:** 6
**Confidence:** 4

**Summary:**

This paper proposes FreqBlender, a new method to generate pseudo-fake faces that effectively simulate the frequency distribution of wild fake faces. Unlike common blending techniques done in the spatial domain, their method blends frequency knowledge. An analysis is conducted, showing that three frequency components are present in faces, namely semantic information, structural information, and noise information. They demonstrated that structural information contains forgery traces.

To this end, the first stage of their method employs FPNet, a novel architecture built to disentangle the input fake face into the three different frequency components. As no ground truth exists for this task, carefully crafted objectives provide the necessary supervision.

In the second stage, the trained network is used to parse the frequency components, and the structural component is extracted from a given fake face. It is then blended into a real face to obtain the pseudo-fake.

The method outperforms the state-of-the-art across different relevant datasets.

**Strengths:**

Originality:
This method is the first to mine frequency structural information and propose a way of generating pseudo-fake faces that mimic the frequency distribution of fake faces.

Quality:
The method is well evaluated on different datasets, and the results are consistent.

Clarity:
The paper is well written, and the method is well explained. Figures, along with a preliminary analysis of the frequency components present in faces, are provided, which help to understand the method.

Significance:
This method provides a new pseudo-fake mechanism that complements spatial blending, which is the reference in the literature. The authors have shown that their method improves the results of different frame and spatial blending-based methods such as DSP-FWA, I2G, Face-Xray, and SBI. The authors claim that the code will be released in the future. To ease the reproduction and adoption of this technique, we encourage them to also release the pretrained weights of FPNet.

**Weaknesses:**

References Missing:
Some references are missing, for example, methods that are frame-based and only use real faces during training [1, 2], as well as other recent detectors [3, 4].

Additional Overhead:
During the training of the detector, the inference of FPNnet (1 encoder and 3 decoders) is required for generating a pseudo-fake. This adds an overhead during training. The authors should include an efficiency analysis.

[1] Li et al., Pixel bleach network for detecting face forgery under
compression, in IEEE Transactions on Multimedia 2023
[2] Larue et al., SeeABLE: Soft Discrepancies and Bounded Contrastive Learning
for Exposing Deepfakes, in ICCV 2023
[3] Dong et al., Implicit Identity Leakage: The Stumbling Block to Improving Deepfake Detection Generalization, in CVPR 2023
[4] Guo et al., AltFreezing for More General Video Face Forgery Detection, in CVPR 2023

**Questions:**

Clarification Needed on FPNet Training:
While the authors explain that for deepfake detection, pseudo-fakes generated using SBI are used as a substitute for fake faces from FF++, the authors should include the results when real fake faces from FF++ are used in Table 1.

Use of PixelShuffle:
Why is a PixelShuffle used in the decoder of FPNet?

FPNet Direct Application:
Can the FPNet be used directly for deepfake detection? What happens when a real face is input into the FPNet?

Ablation Study and Frequency Knowledge:
The authors successfully conduct an ablation study on the backbone (Table 5) and show that results improve consistently when compared to SBI. Why does this method introduce frequency knowledge when all the tested backbones are spatial (e.g., EfficientNet-b4)? It would be interesting to compare the proposed method with a built-in frequency-based backbone (e.g., Face Forgery Network (F3-Net) [3], AFFGM [1], or the multi-scale high-frequency feature extractor from [2]).

Results on Highly Compressed Data:
What are the results of the method on highly compressed data, i.e., FF++ LQ?

[1] Li et al., Frequency-aware Discriminative Feature Learning Supervised by Single-Center Loss for Face Forgery Detection, in CVPR 2021
[2] Luo et al., Generalizing Face Forgery Detection with High-frequency Features, in  CVPR 2021
[3] Thinking in Frequency: Face Forgery Detection by Mining Frequency-aware Clues, in ECCV 2020

**Limitations:**

Acknowledgement of Limitations:
The authors acknowledge the usual limitations of pseudo-fake generation methods, i.e., the hypothesis that the test face is crafted using face-swapping techniques may not hold when the test face is generated using other techniques.

Discussion on Societal Impact:
The race between attackers and defenders is a well-known issue in the field of deepfake detection. The authors should discuss the potential negative societal impact of their work and how it could be exploited by attackers to generate more realistic deepfakes by simply training their generators to fool the proposed method.

---

> ### Author Rebuttal · Authors · 2024-08-07
>
> We appreciate the positive feedback regarding the originality, quality, clarity, and significance of our work, and are grateful to the constructive comments.
>
> **Q1: Some references are missing, e.g., [1, 2, 3, 4].**
>
> **R1:** Thanks for the suggestion. We will include these related references in the revision.
>
> **Q2: Additional Overhead: FPNet (1 encoder and 3 decoders) adds an overhead during training. The authors should include an efficiency analysis.**
>
> **R2:** Thanks for the constructive suggestion. Note that the encoder only contains four convolutional layers and each decoder  contains four layers made up of a convolutional layer and a PixelShuffle operation (L173-L175). Thus, FPNet is lightweight and has small overhead. **Table A shows the efficiency analysis regarding FLOPs, Params and Run-time of FPNet**. Using an Nvidia 3080ti GPU, the run-time is 248 FPS (0.004 seconds per image), and creating a pseudo-fake face costs 0.074 seconds on average. Thus our method only introduces minimal overhead in training detectors.
>
> **Table A: Efficiency analysis of FPNet.**
> | Architecture|  Params (M)  | FLOPs(G)| Run-time (FPS) |
> |:-----------:|:--------:|:-----:|:-----:|
> | FPNet (Encoder * 1,Decoder * 3)     |   20.14  | 29.10 |  248  |
>
> **Q3: While the authors explain that pseudo-fakes generated using SBI are used as a substitute for fake faces from FF++, the authors should include the results when real fake faces from FF++ are used in Table 1.**
>
> **R3:** Thanks for the constructive suggestion. We conduct an extra experiment accordingly, as shown in **Table B**. It can be seen that by using "real" fake faces to create pesudo-fakes with our method, the performance is notably degraded across all datasets. This is because compared to SBI, only using the fake faces from FF++ lack diversity, primarily reflecting the frequency distribution of FF++ instead of real-world fake faces. We will include these results into revised version.
>
> **Table B: Performance of using fake faces from FF++.**
> |     Method  | Type |   CDF | DFDC | DFDCP | FFIW  |
> |:-----------:|:----:|:-----:|:----:|:-----:|:-----:|
> | FreqBlender |"real" fake |   75.79    |  66.30   |   82.01   |  67.70 |
> | FreqBlender |SBI | 94.59 | 74.59| 87.56 | 86.14 |
>
> **Q4: Why is a PixelShuffle used in the decoder of FPNet?**
>
> **R4:** PixelShuffle is an effective upsampling operation that is widely used in generative models. This operation rearranges the channels and reshapes the features by a specified factor. Since the decoder is designed to generate frequency component masks, we employ PixelShuffle operations for upsampling. We will include these descriptions in the revision for better clarity.
>
>
> **Q5: Can the FPNet be used directly for deepfake detection? What happens when a real face is input into the FPNet?**
>
> **R5:** Thanks for the thoughtful questions. The responses are as follows:
> 1) **FPNet is not intended for direct deepfake detection**. This is because that FPNet is designed to analyze the frequency components, which serves as a preprocessing step to create effective pseudo-fake faces for training deepfake detectors.
> 2) **FPNet can decompose the frequency domain into three components for both real and fake faces.** But for real faces, the frequency component corresponding to structural information does not contain forgery traces.
>
> **Q6: ITable 5) Why does this method introduce frequency knowledge when all the tested backbones are spatial (e.g., EfficientNet)?**
>
> **R6:** Thanks for the thoughtful question. Our method creates effective pseudo-fake faces by blending frequency knowledge. These pseudo-fake faces contain essential frequency knowledge and are used for training deepfake detectors. Thus, the frequency knowledge can be introduced, even though the backbones are spatial.
>
> **Q7: (Table 5) It would be interesting to compare the proposed method with a built-in frequency-based backbone (e.g., F3-Net [3], AFFGM [1], or the multi-scale high-frequency feature extractor from [2]).**
>
> **R7:** Thanks for the constructive suggestion. As suggested, we attempt to explore the effect of the backbones in frequency-based methods. Since AFFGM [1] has not released its codes, we are unable to test our method on this backbone. Nevertheless, we retrain F3-Net [3] and GFFD [2] using their released codes. The results, **shown in Table C**, indicate that the performance of these frequency-based backbones also improved, demonstrating the effectiveness of our method on frequency-based backbones. We will include these results and analysis in the revision.
>
> **Table C: Performance of our method with F3-Net [3] and GFFD [2].**
> |  Method          | CDF   | FF++ | DFDCP | FFIW  | Avg   |
> |:-----------:     |:-----:|:----:|:-----:|:-----:|:-----:|
> | F3-Net [3] + SBI | 84.94 | 93.42| 79.29 | 73.42 | 82.77 |
> | F3-Net [3] + Ours| 88.10 | 95.16| 84.32 | 74.49 | 85.52 |
> | GFFD [2] + SBI   | 81.34 | 91.81| 77.19 | 65.53 | 78.97 |
> | GFFD [2] + Ours  | 86.71 | 92.18| 78.25 | 77.45 | 83.65 |
>
>
> **Q8: Results on Highly Compressed Data: What are the results on highly compressed data, i.e., FF++ LQ?**
>
> **R8:** Thanks for the insightful question. **Table D** shows the results of our method when testing on FF++ LQ set. It can be observed that the performance of all methods significantly declines on the LQ set, which aligns with our expectations since compression operations can obscure forgery traces, making detection more challenging. However, our method still achieves the best performance compared to others, demonstrating better generalization ability on low-quality videos.
>
> **Table D: Performance of our method on FF++ LQ.**
> |    Method   | FF++ LQ  |
> |:-----------:|:-----:   |
> | I2G         |  52.20   |
> | Face x-ray  |  65.41   |
> | SBI         |  76.11   |
> | FreqBlender |  77.56   |
>
>
> **Q9: We encourage them to also release the pretrained weights of FPNet**
>
> **R9:** Thanks for the suggestion. We will release the weights along with the codes after acceptance.

---

> ### Author Response · Authors · 2024-08-12
>
> Dear Reviewer Xqko,
>
> Thank you once again for your insightful comments! We look forward to receiving your feedback. We hope the new experiments and additional explanations have demonstrated the merits of this paper. If you have any further questions, please do not hesitate to reach out.
>
> Best regards, Authors

---

### Official Review · Reviewer_RzXt · 2024-07-13

**Soundness:** 3
**Presentation:** 3
**Contribution:** 3
**Rating:** 5
**Confidence:** 5

**Summary:**

This paper have introduced an effective way to improve the generalization of DeepFake detection via generating pseudo-fake faces by blending frequency knowledge.  The proposed approach achieves state-of-the-art (SOTA) results on various deepfake detecion datasets.

**Strengths:**

1) This paper attempts to combine frequency domain information and spatial domain information to deal with deepfake detection, which is very interesting. Spatial domain blending is very common, but it is still very rare to use it in the frequency domain, which is quite innovative.

2) The writing of this paper is easy to understand and the logic is clear.

3) The experiments in this paper are sufficient and effectively support the author's theoretical basis.

**Weaknesses:**

1) The idea of ​​this paper is similar to some already published papers, such as [1], [2] and [3]. I hope this paper can cite and further analyze them:

[1] Tan, C., Zhao, Y., Wei, S., Gu, G., Liu, P., & Wei, Y. (2024, March). Frequency-Aware Deepfake Detection: Improving Generalizability through Frequency Space Domain Learning. In Proceedings of the AAAI Conference on Artificial Intelligence (Vol. 38, No. 5, pp. 5052-5060).

[2] Yu, B., Li, W., Li, X., Lu, J., & Zhou, J. (2021). Frequency-aware spatiotemporal transformers for video inpainting detection. In Proceedings of the IEEE/CVF International Conference on Computer Vision (pp. 8188-8197).

[3] Frank, J., Eisenhofer, T., Schönherr, L., Fischer, A., Kolossa, D., & Holz, T. (2020, November). Leveraging frequency analysis for deep fake image recognition. In International conference on machine learning (pp. 3247-3258). PMLR.

2) Moreover, Ref.30 is an important and widely cited work that introduces the frequency domain into deepfake detection. I don’t quite understand why the author does not compare and analyze it with this work.

3) The source information, page numbers, publishers, etc. of many references are incomplete, and some even are incorrect. For example, [4] should come from CVPR2024 instead of arXiv.

[4] Choi, J., Kim, T., Jeong, Y., Baek, S., & Choi, J. (2024). Exploiting Style Latent Flows for Generalizing Deepfake Video Detection. In Proceedings of the IEEE/CVF Conference on Computer Vision and Pattern Recognition (pp. 1133-1143).

4) The faces in Figure 6 should have been resized. It would be better if the author could show the original resolution image to help readers better understand the experimental results.

5)  Why is the fluctuation of  λ3 so much larger than that of other parameters? I hope the author can provide the corresponding theoretical basis and further detailed analysis.

6) If the author can answer and revise the relevant questions in the final version, I will consider increasing the final score in the next round.

**Questions:**

Some images in the paper will be a little blurry after zooming in, so it is best to convert all images to pdf or eps format.

**Limitations:**

Yes.

---

> ### Author Rebuttal · Authors · 2024-08-07
>
> We sincerely thank the reviewer for the positive comments on the novelty, writting and experiment configuration, and for the constructive suggestions.
>
> **Q1: The idea of this paper is similar to some already published papers, such as [1], [2] and [3]. I hope this paper can cite and further analyze them.**
>
> **R1:** Thanks for highlighting these references. We analyze them in the following:
> * **Analysis of [1][3]**: References [1] and [3] focus on detecting deepfakes (e.g., GAN-generated faces) by learning frequency information. Specifically, [1] introduces a dedicated architecture to extract frequency information hidden in deepfake faces, while [3] directly analyzes the frequency spectrum of these faces. **Our method differs significantly from these methods:**
>     * **Motivation and methodology are different**: [1][3] focus on designing frequency-sensitive architectures or strategies to capture frequency information from input samples. In contrast, our method creates pseudo-fake faces for training deepfake detectors. This is achieved by a novel frequency-blending strategy to make pseduo-fake faces resemble the real-world fake faces.
>     * **Scope is different**: It should be noted that [1][3] mainly focus on detecting whole face image synthesis, whereas our method lies in the scope of face-swap deepfake detection.
> * **Analysis of [2]**: Reference [2], on the other hand, explores the use of frequency knowledge for **general image forensic tasks (e.g., object inpainting), rather than deepfake face detection**. Thus, this method could not be directly adapted to our task.
>
> In summary, while these references also employ frequency information, the motivation, methodology and scope are different. We will include these papers and their analyses in the revision.
>
> [1] Frequency-Aware Deepfake Detection: Improving Generalizability through Frequency Space Domain Learning. AAAI 2024.
> [2] Frequency-aware spatiotemporal transformers for video inpainting detection. ICCV 2021.
> [3] Leveraging frequency analysis for deep fake image recognition. ICML 2020.
>
>
> **Q2: [30] is an important and widely cited work that introduces the frequency domain into deepfake detection. Why the author does not compare and analyze it with this work.**
>
> **R2:** Thank you for the question. Reference [30] is an earlier work (ECCV 2020) that investigated the use of the frequency domain. We cited this work in our main text in order to provide context for using DCT (L108). Since this approach is somewhat outdated, we did not include it in Table 1 of the main text. As suggested, we retrain the model using the released code and evaluate it rigorously according to their instructions. The results are shown in **Table A**. It can be seen that [30] can only perform decently on these recent datasets. We will include this analysis in the revised version.
>
> **Table A: Performance of [30] and our method.**
> |   Method    | CDF | DFDC | DFDCP | FFIW  |
> |:-----------:|:-----:|:----:|:-----:|:-----:|
> | [30]        | 72.93 | 61.16| 81.96 | 61.58 |
> | FreqBlender | 94.59 | 74.59| 87.56 | 86.14 |
>
> **Q3: Why is the fluctuation of λ3 so much larger than that of other parameters? I hope the author can provide the corresponding theoretical basis and further detailed analysis.**
>
> **R3:** Thanks for the constructive suggestion. $\lambda_3$ is the coefficient of Quality-agnostic Loss (L222), which helps regulate the intensity of noise information. Since Prior and Integrity Loss also play a role in controlling noise intensity (implicitly), $\lambda_3$ should be relatively subtle compared to other coefficients. To furthter illustrate this trend, we conduct an additonal experiment with $\lambda_3 = 0.1$. Note that $\lambda_3 = 1,0.01,0.001$ has been studied in Table 6 of Supplementary. The results are shown in **Table B**, indicating that our method is slightly improved with a lower $\lambda_3$. We will provide a more detailed analysis in the revision.
>
> **Table B: Effect of our method on different loss proportions.**
> | $\lambda_1$ | $\lambda_2$ | $\lambda_3$ | $\lambda_4$ | CDF | FF++ | DFDCP | FFIW  | Avg |
> |:-----------:|:----:|:----:|:-----:|:----:|:-----:|:-----:|:-----:|:-----:|
> |0.1 | 1 | 0.1  | 0.5       | 93.59 | 95.60 | 86.60 | 84.78 | 90.14 |
> |0.1 | 1 | 0.01 | 0.5       | 94.27 | 96.11 | 87.81 | 85.54 | 90.93 |
>
> **Q4: Many references need to be revised.**
>
> **R4:** We will carefully check the references and revise them accordingly.
>
> **Q5: It would be better if the author could show the original resolution image (Figure 6) to help readers better understand the experimental results.**
>
> **R5:** We will revise Figure 6 accordingly.
>
>
> **Q6: Some images in the paper are a little blurry after zooming in. It is best to convert all images to pdf or eps format.**
>
> **R6**: We will update these figures accordingly to enhance clarity.

---

> ### Author Response · Authors · 2024-08-12
>
> Dear Reviewer RzXt,
>
> We sincerely appreciate your time and thoughtful comments. We eagerly look forward to your feedback, especially on the issues we've addressed in our rebuttal. Our main goal is to ensure that our response aligns closely with your suggestions. Your input is invaluable to improving our work.
>
> Best regards, Authors

---

### Author Rebuttal · Authors · 2024-08-07

We sincerely thank all reviewers for their valuable time and professional comments. We are encouraged by the positive feedback and suggestions on the following aspects:
* **Soundness, Presentation, and Contribution**:
    * **All reviewers** rate these sections as **Good** in their reviews.
* **Novelty of our method**:
    * **Reviewer RzXt** acknowledges that our method is quite interesting and innovative in terms of frequency blending.
    * **Reviewer Xqko** recognizes the originality and significance of our method, particularly in the new pseudo-fake mechanism.
    * **Reviewer Z1eA** highlights that the method addresses an existing challenge by taking an innovative approach of frequency blending.
* **Extensive and sufficient experiments**:
    * **Reviewer RzXt** notes that the experiments are sufficient and effectively support the theoretical basis.
    * **Reviewer Xqko** comments that our method is well evaluated, with consistent results.
    * **Reviewer Z1eA** emphasizes that our method is valdate across various DeepFake datasets with different backbones.
* **Well-written and clear logic**:
    * **Reviewer RzXt**, **Reviewer Xqko** and **Reviewer Z1eA** comment that the paper is well-written, the logic is clear and the method is well explained.
    * **Reviewer Xqko**, **Reviewer GPf8** like the preliminary analysis in figures of frequency components in faces.

We appreciate that **Reviewer GPf8** has a positive view of our  unsupervised way of learning frequency comopents, the analysis of frequency distribtuion and the problem formation along with learning objects. Meanwhile, **Reviewer GPf8** also raises two concerns: **"...but I am inclined to reject it because it does not achieve reasonable sota performance and does not show generalization to diffusion-generated images."**

We would like to highlight the responses to these concerns:
* **Concern of SOTA**: We would like to highlight that **our method achieves the highest number of top-1 rankings compared to all others (best performance on 3 out of 4 datasets)**, and ranks 3rd on the DFDC dataset among the 20 methods compared. Given the wide variety of detection strategies and deepfake datasets, we believe this achievement can demonstrate the comprehensive superiority of our method. **Moreover, we respectfully believe that both the AC and all reviewers could agree with that, the value and contribution of a work should not solely be judged by empirical results, but also by the innovative insights it offers for future research.**
* **Concern of generalization to Diffusion-generated images**: We would like to thank the reviewer for this thoughful comment.
    * **Our method is in scope of face-swap deepfake detection**: As described in Introduction (L29-L31) and stated in the limitation section (L322-L325), our method targets for the face-swap deepfake detection, **a significant topic in recent years, as evidenced by works such as Face x-ray [14] (CVPR2020), PCL [15] (ICCV2021), SBI [16] (CVPR2022), UCF [12] (ICCV2023), BiG-Arts [25] (PR2023), F-G [43] (CVPR2024), and LSDA [23] (CVPR2024)**. This scope limitation has been noted by other reviewers as well.
    * **It is important to note that detecting diffusion-generated faces and face-swap deepfake are typically two different tasks**: Detecting Diffusion-generated images typically falls under the category of whole image synthesis, as seen in works such as DIRE (ICCV2023, the suggested work by reviewer), AVG (ICASSP2023). In these works, they do not validate themselves on face-swap deepfake datasets studied in our paper. Therefore, **while detecting diffusion-generated images is also an important task, it falls outside the scope of our method.**
    * **In response to the suggestion, we investigate whether our method can facilitate the detection of Diffusion-based generated images.** We conduct an additional scenario: **Diffusion-based face-swap deepfake detection**, where the diffusion model is adapted to a face-swap scenario. In this scenario, a diffusion model (Collaborative Diffusion (CVPR2023)) is used to synthesize faces, which are then blended into original videos. The results are shown in **R3** of our response to **Reviewer GPf8**, demonstrating the efficacy of our method in detecting Diffusion-based face-swap deepfakes. We will include this experiment in the revision.

We hope above responses could address the concerns of **Reviewer GPf8**, and facilitate AC to make a more comprehensive assessment of the value and contribution of our work.

---

### Decision · Program_Chairs · 2024-09-25

**Decision:**

Accept (poster)

**Comment:**

This paper received mixed reviews, ranging from "Borderline reject" to "Weak Accept". Reviewers acknowledged the innovative DeepFake detection method proposed in the paper (Z1eA, Xqko, RzXt). With Xqko stating that "This method is the first to mine frequency structural information and propose a way of generating pseudo-fake faces that mimic the frequency distribution of fake faces" (Xqko), and noted the analysis it provided (Xqko, GPf8) and quality of writing (RzXt, Xqko).

The reviews raised a number of concerns, including limited novelty compared to published work that was not all referenced (RzXt, Xqko). The reviewers were concerned with the experiments, noting it did not always achieve state-of-the-art detection (GPf8) and the limited evaluation against recent DeepFake detection methods and datasets, particularly those using diffusion models (GPf8, Z1eA).

The authors provided detailed responses, though the recommendations ultimately remained divergent. I see the remaining concerns as insufficient reasons for a rejection, preferring to err on the side of acceptance in such cases.